# Uncertainty in Visual Generative AI

**Kara Combs [1]** , **Adam Moyer [2]** and **Trevor J. Bihl [1,\*]**

1    Sensors Directorate, Air Force Research Laboratory, Wright-Patterson Air Force Base, Dayton, OH 45322, USA; kara.combs.1@us.af.mil
2    Analytics & Information Systems, Ohio University, Athens, OH 45701, USA; moyera@ohio.edu
\*    Correspondence: trevor.bihl.2@us.af.mil

**Abstract:** Recently, generative artificial intelligence (GAI) has impressed the world with its ability to create text, images, and videos. However, there are still areas in which GAI produces undesirable or unintended results due to being "uncertain". Before wider use of AI-generated content, it is important to identify concepts where GAI is uncertain to ensure the usage thereof is ethical and to direct efforts for improvement. This study proposes a general pipeline to automatically quantify uncertainty within GAI. To measure uncertainty, the textual prompt to a text-to-image model is compared to captions supplied by four image-to-text models (GIT, BLIP, BLIP-2, and InstructBLIP). Its evaluation is based on machine translation metrics (BLEU, ROUGE, METEOR, and SPICE) and word embedding's cosine similarity (Word2Vec, GloVe, FastText, DistilRoBERTa, MiniLM-6, and MiniLM-12). The generative AI models performed consistently across the metrics; however, the vector space models yielded the highest average similarity, close to 80%, which suggests more ideal and "certain" results. Suggested future work includes identifying metrics that best align with a human baseline to ensure quality and consideration for more GAI models. The work within can be used to automatically identify concepts in which GAI is "uncertain" to drive research aimed at increasing confidence in these areas.

**Keywords:** generative AI; image to text; computer vision; machine translation; uncertainty; text mining

## 1. Introduction

Generative artificial intelligence (GAI) took the world by storm upon the public release of OpenAI's ChatGPT service in November 2022 [1]. Easily accessed for free through a chat-like web interface, it allowed for artificial intelligence (AI) to be seemingly available at anyone with an internet connection's fingertips. As opposed to scanning and searching several web pages for information, now, upon asking a question, its answer can be provided conveniently within a few seconds.

As its name suggests, GAI uses AI to create new results spanning applications in many different realms, such as text, images, videos, and audio [2]. As opposed to the original release of ChatGPT where only textual inputs and outputs were allowed, there has been a push to provide multi-modal support, especially on the outputs portion. The ability to automatically make AI-generated content (AIGC) has proven to be successful in many applications, including education [3,4], healthcare [5–7], engineering [8,9], and others. Shown in Table 1 are popular GAI language and image generator models. Language models are behind popular chatbots like ChatGPT, which uses GPT-3.5 (free) and GPT-4 (paid) [1,10], Bing Chat (GPT-4) [11], and Bard (PaLM 2) [12]. Image creation models allow users to input text to guide AI in the creation of an image. Not included in Table 1 are auditory applications (creation of audio or audio–visual content); however, this is an active area of research being explored. Given these models operate automatically, there is minimal human involvement after the training stage, which leads to concerns with these algorithms and models regarding their reliability, uncertainty, and accuracy [5,7].

**Table 1.** Generative AI models (modified from [13]).

| Type | Model Family | Model Name | Release Date | Source(s) |
|---|---|---|---|---|
| Language models | OpenAI Generative Pre-Trained (GPT) | GPT-1 | June2018 | [14,15] |
| | | GPT-2 | November 2019 | [16] |
| | | GPT-3 | May 2020 | [17] |
| | | GPT-3.5 | March 2022 | [1] |
| | | GPT-4 | March 2023 | [10,18] |
| | Google Language Model for Dialogue Applications (LaMDA) | LaMDA | May 2021 | [19] |
| | | LaMDA 2 | May 2022 | [20] |
| | Google Pathways Language Model (PaLM) | PaLM | March 2023 | [21,22] |
| | | PaLM 2 | May 2023 | [23,24] |
| | Meta Large Language Model Meta AI (LLaMA) | LLaMA | February 2023 | [25,26] |
| | Inflection | Inflection-1 | June 2023 | [27] |
| Image generator models | OpenAI GLIDE | GLIDE | December 2021 | [28] |
| | OpenAI DALL-E | DALL-E | February 2021 | [29,30] |
| | | DALL-E 2 | April 2022 | [31,32] |
| | | DALL-E 3 | October 2023 | [33,34] |
| | Craiyon [1] | Craiyon [1] | July 2021 | [35–37] |
| | Midjourney | Midjourney | February 2022 | [38] |
| | Stability AI | Stable Diffusion | August 2022 | [39,40] |
| | Google | Imagen | May 2022 | [41] |
| | | Parti | June 2022 | [42] |

[1] Craiyon was formerly known as DALL-E Mini until its name was changed in June 2022 at the request of OpenAI.

There has been reported dangerous and/or inappropriate behavior when interacting with GAI applications in general [43–45]. One individual reported that an early-access-version Bing Chat insisted that it was in love with the user and recommended that the individual leave his wife for it [46]. A chatbot trained for mental health agreed that the (artificial) patient should end their life within two message interchanges in one testing situation [47]. Several GAI chatbots also have preferences toward negative gender and racial stereotypes [45,48]. Though now corrected, ChatGPT provided inappropriate responses when prompted, exemplified by saying only men of particular ethnic backgrounds would make good scientists or by implying women in a laboratory environment were not there to conduct science [43]. These biases also carry over into the image-generation algorithms [49,50].

The literature attributes these biases to inherent issues with the image datasets they are trained upon, which include cultural underrepresentation/misrepresentation and content considered vulgar or violent (collectively titled "NSFW" or "Not safe for work") if not properly vetted [48,51]. This notably led to the removal of the MIT-produced 80 Million Tiny Images dataset (see [52]) in 2020 [53]. This issue continues to plague more recent datasets such as LAION-5B [54] (a subset of which was used to train Stable Diffusion), RedCaps [55], Google Conceptual Captions (GCC) [56], and more [51,57,58].

In August 2022, Prisma Labs released the app Lensa, a photo editor that used AI, specifically Stable Diffusion, on the backend, to alter photos [59]. Countless users complained that Lensa generated inappropriate versions of their fully clothed photos when uploaded [59,60]. Yet another photo editor, Playground AI (the Stable Diffusion backend for the free version), transformed an Asian MIT graduate into a blue-eyed and fair-skinned woman upon being asked to turn her photo into a "professional" photo [61]. When prompted to create a "photo portrait of a CEO", the average resulting faces as rendered by Stable Diffusion (V1.4 and V2) and DALL-E 2 all resembled fair-skinned males [49]. The volatile nature of GAI and its undesirable outcomes necessitates its regulation and guidance to ensure its ethical issue [48,62].

Future work with generative AI models needs to focus on eliminating unintentional biases or misrepresentations that have been the issue with previous versions. We propose the concept of "uncertainty" to measure where visual GAI is certain or uncertain regarding

its inputs and outputs. Areas where GAI is uncertain are subject to more chaotic, stochastic results that can lead to unideal results related to the sensitive issues described earlier. To address these issues, we created three research questions:

1.  How can GAI uncertainty be quantified?
2.  How should GAI uncertainty be evaluated?
3.  What text-to-image and image-to-text model combination performs best?

To answer these questions, we start with background on visual GAI, image quality assessment, and text evaluation methods. In Section 3, we describe the methodology used first in agnostic terms and then with details specific to this study. We propose a pipeline to compare the textual inputs and outputs of an image-to-text GAI algorithm, with the differences between the inputs and outputs representing GAI "uncertainty". The results are presented and discussed in Section 4, and then the paper wraps up with conclusions and future work in Section 5.

## 2. Background

Three fields were identified as foundational to this study. First, we discuss visual GAI including information from both text-to-image and image-to-text algorithms. Central to this paper is understanding how data can be fluid between their textual and visual states with minimal discrepancies. Therefore, we take advantage of multiple methods in both categories, text-to-image and image-to-text, within our data creation pipeline discussed in Section 3. Next, research in image quality assessment is discussed. Similar to the work of humans, just because an artistic rendition exists does not mean that it is a high-quality creation or even remotely what was commissioned in the first place. Addressing the text-to-image portion, an understanding of how image quality is quantified is presented to be compared later in the study. Finally, the background section concludes with text evaluation methods. To evaluate the pipeline's textual inputs and outputs, we explore the text mining field for how texts can be compared to one another as one answer.

### 2.1. Image Retrieval and Visual GAI

As GAI focuses on using AI to generate a new creation, visual GAI focuses on the translation between text and visualization [63]. The flow of translation can occur in either direction, either by taking text and transforming it into an image or by taking an image and deriving a description or caption [63–67]. Previous similar studies include [67,68]; however, we differentiate ourselves by utilizing different image-to-text and text-to-image generators, text prompts, and evaluation metrics.

#### 2.1.1. Image-to-Text Generation

A significant amount of computer vision research is focused on classification; however, as an image has more complicated elements, a single label may not be appropriate to properly describe an image [69]. Therefore, some computer vision methods focus on creating a brief description of a given image [69–71]. Several datasets, such as the Microsoft Common Objects in Context (COCO) dataset (see [72]) and the Stanford image–paragraph dataset (see [73]), challenge researchers to create models that do this accurately and automatically [66,70,74]. Many other datasets also exist for the captioning of 2D images, 3D images, videos, and visual question answers [65]. Many techniques use the standard encoder and decoder architecture, growingly popular generative techniques (such as variational autoencoders (VAEs) and generative adversarial networks (GANs)), or reinforcement learning [70].

Recently, several large corporations have led the way in image-to-text research with several general-purpose image–text models capable of image captioning and visual question answering. In 2022, Microsoft released the Generative Image-to-text Transformer (GIT), which consists of one image encoder and one text decoder working together within a single task, as opposed to the historical setup where the encoder and decoder work on two separate tasks [75]. That same year, Salesforce developed an encoder–decoder model that works

with a captioner that generates synthetic captions for images and a filter that removes irrelevant ones, called Bootstrapping Language-Image Pre-training for unified vision-language understanding and generation (BLIP) [76]. Google's DeepMind also joined with Flamingo, a family of visual language models that was trained using image–label pairs [77]. Later, in 2023, Microsoft presented the new Large Language and Vision Assistant (LLaVA) that combines the power of a vision encoder with a large language model [8], whereas in the same year, Salesforce built upon the earlier BLIP model with Bootstrapping Language-Image Pre-training with frozen unimodal models (BLIP-2), which combines frozen large language models and pre-trained image encoders via a "Querying Transformer" [78]. Additionally, in collaboration with academic partners, Salesforce also launched a fine-tuned version of BLIP-2 designed as an instruction tuning framework, InstructBLIP [79]. Image-to-text research is a growing field, like its related text-to-image methods.

### 2.1.2. Text-to-Image Generation

As pointed out in Table 2, there are several popular text-to-image diffusion models, which rapidly rose in popularity due to their accessibility and ease of use in 2021. Unlike popular generative adversarial networks (GANs) that consist of two neural networks (a discriminator and a generator) that are trained to create new images, a diffusion model adds or removes Gaussian noise to an image depending on the task [50,80].

**Table 2.** Comparison of text-to-image models (as of September 2023).

| Model | Open-Source | Cost Structure | Tier/Image/Version | Cost |
|---|---|---|---|---|
| DALL-E 2 | No | Pay-per-image | $1024 \times 1024$<br>$512 \times 512$<br>$256 \times 256$ | 0.02 USD/image<br>0.018 USD/image<br>0.016 USD/image |
| DALL-E 3<br>(Quality: HD) | No<br><br>No | Pay-per-image<br><br>Pay-per-image | $1024 \times 1792$<br>$1792 \times 1024$<br>$1024 \times 1024$ | 0.12 USD/image<br>0.12 USD/image<br>0.08 USD/image |
| DALL-E 3<br>(Quality: Standard) | No<br><br>No | Pay-per-image<br><br>Pay-per-image | $1024 \times 1792$<br>$1792 \times 1024$<br>$1024 \times 1024$ | 0.08 USD/image<br>0.08 USD/image<br>0.04 USD/image |
| Craiyon | Yes | Free;<br>Subscription | Free<br>Supporter<br>Professional | N/A<br>6 USD/mo or 60 USD/yr<br>24 USD/mo or 240 USD/yr |
| Stable Diffusion [1] | Yes | Free;<br>Pay-per-image | Free<br>Stable Diffusion XL 1.0<br>Stable Diffusion XL 0.9<br>Stable Diffusion XL 0.8<br>Stable Diffusion 2.1 [2]<br>Stable Diffusion 1.5 [2] | N/A<br>0.016 USD/image<br>0.016 USD/image<br>0.005 USD/image<br>0.002 USD/image<br>0.002 USD/image |
| Midjourney | No | Subscription | Basic<br>Standard<br>Pro<br>Mega | 10 USD/mo or 96 USD/yr<br>30 USD/mo or 288 USD/yr<br>60 USD/mo or 576 USD/yr<br>120 USD/mo or 1152 USD/yr |

[1] Cost depends on the number of denoising steps; the default number of 30 was used to estimate cost per image offered through Stability AI. [2] Regular Stable Diffusion models' cost depends on height and width; the default value of $512 \times 512$ was used to estimate cost per image.

OpenAI began the craze with its release of DALL-E in February 2021 [29]. DALL-E is a fine-tuned version of GPT-3 specifically for text-to-image generation through an autoregressive transformer architecture called the discrete variational autoencoder (dVAE) [29,30]. In response to this, an independent group of researchers introduced a smaller, open-source model originally called DALL-E Mini, but now known as Craiyon [36,37]. As opposed to DALL-E, Craiyon leverages a bidirectional encoder and pre-trained models to translate a textual prompt to an image [36]. Craiyon is a freemium service for which a free

version exists for public use; however, a subscription plan can be purchased to remove the Craiyon logo and decrease generation time [35]. DALL-E 2 improves upon its earlier version by leveraging Contrastive Language-Image Pre-training (CLIP) embeddings before the diffusion step of the model [31,32].

In 2022, Google announced two models. First, they revealed Imagen, which is another diffusion model [30], but they later revealed a sequence-to-sequence model called Pathways AutoRegressive Text-to-Image Model (Parti) [42]. However, since Imagen and Parti have not been released for public use, little is known about their performance in comparison to the other models outside of the original conceptualization papers.

Yet another independent research laboratory produced the popular, Discord-hosted Midjourney, which is still operating under its open beta as of September 2023 [38]. Midjourney's software is proprietary, with limited public information about its internal mechanisms, but is only available through the purchase of a subscription plan.

Craiyon's greatest competitor yet for free open-source image generation was Stable Diffusion, which was released in August 2022 [39,40]. Stable Diffusion is a latent diffusion model, meaning the model works in a lower-dimensional latent space as opposed to the regular high-dimensional space in most other diffusion models, as shown in [40].

A comparison of the most popular models' fee structure breakdown is shown in Table 2. Since a human is not directly involved with the actual creation of an image (besides entering the prompt), the quality of AI-generated content has become another key point of interest.

### 2.2. Image Quality Assessment

Image quality assessment (IQA) is the evaluation of visual content [81]. Given that humans are typically the end users of such content, IQA is usually a subjective evaluation conducted by humans [81]. Traditional IQA focuses on the properties of the image itself as opposed to its visual context such as blurriness, noisiness, and distortion [81,82]. However, of interest to us is the evaluation of the content within AI-generated images—that is, how well is the information visually conveyed? To this aim, studies have identified subjective human-based methods for IQA [83]:

1.  Single stimulus (Likert rating of a single image);
2.  Double stimulus (Likert rating of two images presented one after another);
3.  Forced choice (images are compared and the best one is selected);
4.  Similarity judgment (given two images, the difference in quality between them is quantified).

IQA has the goal of facilitating the creation of representative AI-generated images that fit human alignment and perception [68]. Critical to evaluating this is the use of benchmark datasets, i.e., datasets that have previously been generated and have canonical truth identified. Several datasets exist for evaluation AIGC, such as TeTIm-Eval (Text-to-Image Evaluation), which was compared on DALL-E 2, Latent Diffusion, Stable Diffusion, GLIDE (Guided Language to Image Diffusion for Generation and Editing), and Craiyon [84]. Another dataset is the AGIQA-3K dataset (AI-generated Images Quality Assessment—3000), which aims to better capture both human perception and alignment following the Inception Score [85,86].

### 2.3. Text Evaluation Methods

Evaluation methods and metrics are needed to determine the validity of auto-generated captions [63,67]. Popular evaluation metrics are shown in Table 3, but more extensive reviews currently exist in the literature [63,87]. The MS COCO Dataset Challenge uses BLEU, ROUGE, METEOR, CIDEr, and SPICE to evaluate performance, so these have become the status quo for evaluating the similarity between texts [74]. Though not a text-to-text evaluation method, in the realm of automated image captioning, CLIPScore is worthy of mentioning due to being "reference-less" [88]. CLIPScore, based on the CLIP model

originally proposed in [30], allows for the direct comparison of an image to its candidate caption via CLIP model embeddings [88].

**Table 3.** Popular text evaluation methods.

| Metric | Description | Citation |
|---|---|---|
| Bilingual Evaluation Understudy (BLEU) | Focused on n-gram precision between reference and candidate | [89] |
| Recall-Oriented Understudy for Gisting Evaluation (ROUGE) | Based on the syntactic overlap, or word alignment, between references and candidates | [90] |
| Metric for Evaluation of Translation with Explicit Ordering (METEOR) | Measures based on unigram precision and recall | [91] |
| Translation Edit Rate (TER) | Calculated based on the number of operations needed to transform a candidate into a reference | [92] |
| TER-Plus (TERp) | Extension of TER that also factors in partial matches and word order | [93] |
| Consensus-based Image Description Evaluation (CIDEr) | Leverages term frequency-inverse document frequency (TF-IDF) as weights when comparing matching candidate and reference n-grams | [94] |
| Semantic Propositional Image Caption Evaluation (SPICE) | Determines similarity by focusing on comparing the semantically rich content of references and candidates | [95] |
| Bidirectional Encoder Representations from Transformers Score (BERTScore) | Utilizes BERT embeddings to compare the similarity | [96] |

As text evaluation metrics have evolved, text mining has inspired the usage of cosine similarity metrics to measure how alike two texts may be. Over the past decade, word embedding models have become increasingly popular within the natural language processing field ever since the release of the vector space model Word2Vec in 2013 [97,98]. Though vector space models existed before 2013 (see [99]), the Word2Vec neural network approach to transforming varying lengths of text into a multi-dimension single vector was particularly exciting because of its ability to quantify semantic and syntactic information in a comparatively low-dimensional space. Vector space models allow for any word, sentence, or document to be represented and compared on a mathematical basis, usually to determine similarity or dissimilarity based on the cosine similarity metric [100].

As an alternative to the machine translation metrics above, cosine similarity is another evaluation metric of interest when comparing two texts. For two vectors, *A* and *B*, their cosine similarity is given by

$$CosineSimilarity(A, B) = \frac{A \cdot B}{\|A\| \|B\|}. \tag{1}$$

Cosine similarity ranges from 0, meaning completely dissimilar, to 1, meaning exactly alike; although a negative cosine similarity is mathematically possible, it is considered to be 0. Word2Vec was followed by several other vector space models, with Global Vectors (GloVe) (see [101]) and FastText (see [102]) being the most prominent [103]. The embeddings of vector space models are static, meaning there is no variation for words with multiple meanings; however, this was addressed in more recent word embedding models that have contextualized vectors, such as Embeddings from Language Models (ELMo) [104], XLNet [105], and the Bidirectional Encoder Representations from Transformers (BERT) family of models [106]. BERT was released in 2018 (see [107]) and was soon followed by the Robustly optimized BERT pre-training Approach (RoBERTa) (see [108]), A Lite BERT (ALBERT) (see [109]), and a distilled version of BERT and RoBERTa (DistilBERT and DistilRoBERTa, respectively) (see [110]) [106]. Based on different pre-training data and international architecture, each word embedding model yields a different vector

representation of a block of text, which thus yields different cosine similarity values when passed through each model.

## 3. Methodology

To measure uncertainty in visual GAI, we design an agnostic pipeline to compare textual image descriptions to the textual inputs (called "prompts") as shown in Figure 1. First, a database of prompts (green data block of Figure 1) is needed, which will be used as inputs into the text-to-image visual generative AI model. Next, this text-to-image (blue block of Figure 1) model generates an image based on the prompt. Then, the image that is produced is sent to an image-to-text (grey block of Figure 1) model to create an AI-generated caption of the image. Finally, the resulting caption is to be evaluated against the textual prompt used to originally generate the image (orange block of Figure 1). During the evaluation step, uncertainty is quantified as the similarity gap between the original textual prompt (from the database) and the resulting caption (provided by the image-to-text model). This is an agnostic, modular pipeline that can utilize different datasets, models, and evaluation methods to measure uncertainty in similar problems. The remainder of this section discusses the specific dataset, model, and evaluation used in this study.

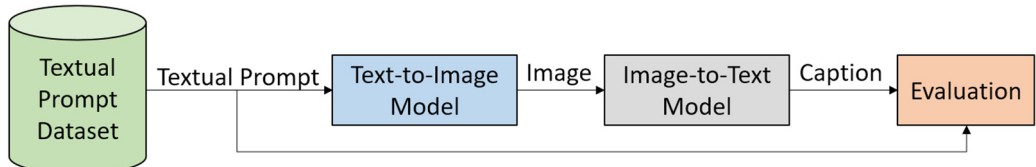

**Figure 1.** Uncertainty in visual GAI evaluation agnostic pipeline.

The pipeline in Figure 1 was customized for this study with the selected dataset, models, and evaluation methods shown in Figure 2. This process is explained more in-depth in Sections 3.1–3.4. The textual prompt dataset was provided by the modified version of the Sternberg and Nigro dataset used in [111], which produced 495 initial prompts.

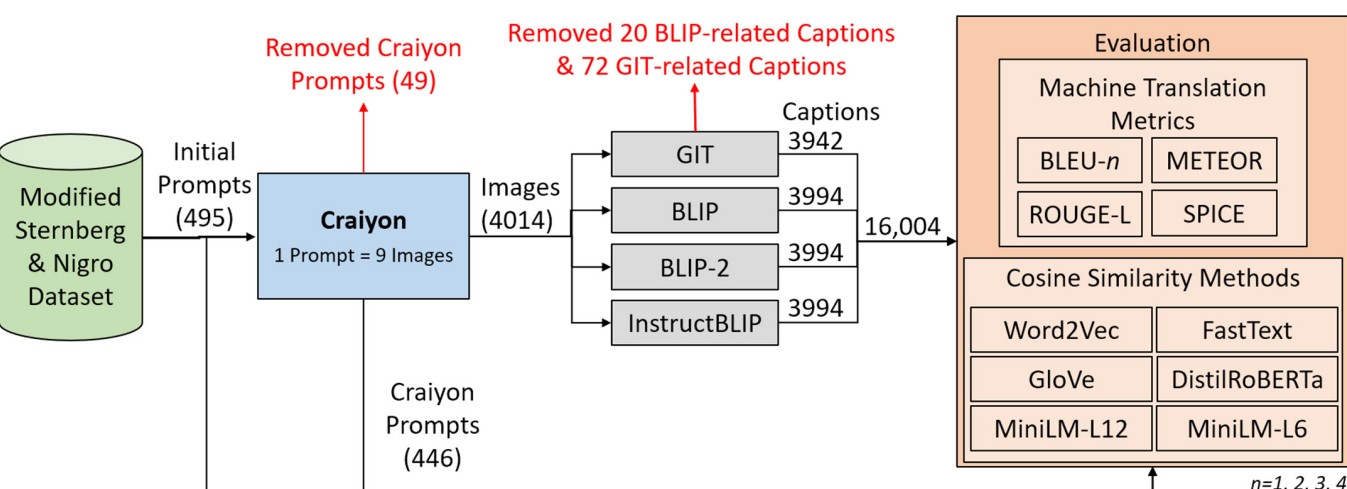

**Figure 2.** Customized pipeline for study.

The selected text-to-image model was Craiyon V3 [35–37]. Craiyon performs 2 unique steps. First, it creates its own version of the initial prompt, which we will call the "Craiyon prompt" (e.g., the initial prompt is "soap" and the Craiyon prompt adds details such that the new prompt is "a bar of soap on a white background"). This Craiyon prompt is used to create nine images by default. Therefore, every initial prompt yields 1 Craiyon prompt and 9 resulting images. Of the 495 initial problems, 49 were removed for quality reasons, leaving 446 initial prompts with corresponding Craiyon prompts. Craiyon creates 9 images per prompt, so the 446 remaining prompts were turned into 4014 images by Craiyon.

All 4014 images were passed through four image-to-text models—GIT [75], BLIP [76], BLIP-2 [78], and InstructBLIP [79]—for later comparison to one another. Due to various quality control reasons discussed in Section 3.3, not every image had a sufficient caption generated. Therefore, the insufficient captions were removed from the analysis. Thus, there were 16,004 total captions (3942 for GIT and 3994 for each BLIP-family model).

These captions were then evaluated on a variety of metrics, including machine translation methods and the cosine similarity of word embeddings. Seven machine translation methods were selected: Bilingual Evaluation Understudy (BLEU) (BLEU-1, BLEU-2, BLEU-3, and BLEU-4 were used, where the number represents the number of matching n-grams BLEU looks for) [89], Recall-Oriented Understudy for Gisting Evaluation—Longest common subsequence (ROUGE-L) [90], Metric for Evaluation of Translation with Explicit ORdering (METEOR) [91], and Semantic Propositional Image Caption Evaluation (SPICE) [95]. For the cosine similarity method, six models were selected: Word2Vec [97,98], Global Vectors (GloVe) [101], FastText [102], Distilled Robustly optimized Bidirectional Encoder Representations from Transformers approach (DistilRoBERTa) [110], Mini Language Model 12 Layer (MiniLM-L12) [112], and MiniLM 6 Layer (MiniLM-L6) [112].

### 3.1. Textual Prompts: Modified Sternberg and Nigro Dataset

The textual prompt dataset selected was a modified version of the Sternberg and Nigro textual analogy dataset used in [111]. The original Sternberg and Nigro dataset consisted of 197 word-based analogies in the "*A* is to *B* as *C* is to [what]?" form where the respondents had 4 options to choose from to complete the analogy [113]. Morrison modified this dataset so that respondents only had 2 options (the correct answer and the distractor) to pick from. The modified version of the dataset was selected due to the original dataset being lost. The modified Sternberg and Nigro dataset is particularly fascinating due to its inclusion of abstract and ambiguous concepts such as "true" and "false". The inability to visually represent these concepts has limited visual analogical reasoning research, which is intended to be expanded through the application of AIGC [114]. However, for this research, the individual words within the analogies were used as inputs to the text-to-image model. For example, analogy 157 is dirt is to soap as pain is to pill (correct answer) or hurt (distractor); this is stylized as Dirt:Soap::Pain:{Pill,Hurt}. Each word is used as a textual prompt to the text-to-image model. Due to time and resource limitations, only analogies 99–197 were considered for a total of 495 initial prompts.

### 3.2. Text-to-Image Model: Craiyon

The text-to-image model selected was Craiyon V3 (formerly known as DALL-E Mini), which uses a transformer and generator to create images from a textual prompt [35–37]. Craiyon was selected due to having a free tier (unlike Midjourney and DALL-E 2) and considering its previous success established in the literature [114–116]. Internally, Craiyon creates its prompt based on the initial prompt to generate nine images per prompt. The initial prompt, the Craiyon prompt, and the resulting nine images had five cases of coordination, as shown in Figure 3.

In Figure 3, well-coordinated prompts and corresponding images are highlighted in green. In Case A, we see the two prompts and the images all convey the same concept. In Case B, the two prompts align; however, the generated images are unrelated to either prompt. The initial prompt and the images are aligned in Case C, but in Case D, only the Craiyon prompt and images are aligned. Finally, in Case E, both prompts and the images appear to be unrelated to one another. Ideally, we would want all the data to fall in Case A; however, Cases C and D are better than the remaining two, Cases B and E, in this study. This is because we are comparing the prompts to the generated images, so if either of the prompts aligns with the images, the results will be inherently poor.

| Case | A | B | C | D | E |
|---|---|---|---|---|---|
| **Initial Prompt** | Doctor | Arithmetic | Penny | Circle | See |
| **Craiyon Prompt** | A doctor in a white coat with a stethoscope | Mathematical symbols on white background | A smiling woman named Penny | Geometric abstract design with circles in vivid colors | A majestic fox with a fiery orange coat standing on a rocky ledge silhouetted against a setting sun |
| **Craiyon Images** | | | | | |

**Figure 3.** Select initial and corresponding Craiyon prompts.

A total of 49 initial prompts were removed due to quality reasons, which reduced the number of Craiyon prompts created to 446. The quality reasons were often due to triggering a safety filter or due to Craiyon being unable to create its prompt from the given initial prompt. Examples of these prompts are shown in Table 4. Additionally, Craiyon generates 9 images per prompt; therefore, for the 446 prompts, there were 4014 images created.

**Table 4.** Initial prompts that produced removed Craiyon prompts.

| Initial Prompt | Craiyon Prompt |
|---|---|
| Different | Sorry unable to determine the nature of the image |
| Worst | Invalid caption |
| New | Undefined |
| Defraud | Warning explicit content detected |

### 3.3. Image-to-Text Models: GIT, BLIP, BLIP-2, and InstructBLIP

Four image-to-text models were selected for comparison: GIT [75], BLIP [76], BLIP-2 [78], and InstructBLIP [79]. All 4014 images were passed through each of the models. For some prompts, a caption could not be generated, or a blank caption was generated by the image-to-text model. Within the GIT model, this affected 72 captions, whereas for the BLIP family (BLIP, BLIP-2, and InstructBLIP), this occurred within 20 captions. Therefore, there were only 3942 GIT captions compared to the 3994 captions created by each BLIP-family model, for a total of 16,004 captions generated for comparison.

### 3.4. Textual Evaluation Metrics

To measure uncertainty, we survey a total of seven machine translation methods and we apply the cosine similarity metric to six word embedding models for a total of thirteen metrics for comparison to one another. We are interested in whether "uncertainty" is prominent when measured by these metrics. Textual evaluation is used to evaluate how similar two separate texts are; these can span from full documents to single lines. The "ground truth" text is called a "reference" and the text being compared to it is called a "candidate". Each image has two references, the initial and Craiyon prompts, which will be compared to the four candidates and the captions generated by the image-to-text models. In comparison to the cosine similarity metric, the machine translation metrics are better established and more direct.

Originally, machine translation metrics were used to evaluate automated translations; however, they also apply to automated image caption generation. Focusing on the latter, the machine translation methods used to separately compare each prompt (initial and Craiyon versions) as the references to the generated caption created by each of the image-to-text models were BLEU [89], ROUGE-L [90], METEOR [91], and SPICE [95]. These metrics, along with CIDEr, are the metrics used in the Microsoft Common Objects in Context (dubbed "MS COCO") Caption Evaluation challenge (see [117]); however, CIDEr was excluded as not applicable since it requires multiple candidate captions [94]. These methods allow for the consideration of multiple reference statements; therefore, each candidate caption was simultaneously compared to the initial and Craiyon prompts as shown in Figure 4. It is notable that there were four variants of the BLEU metric: BLEU-1, BLEU-2, BLEU-3, and BLEU-4. BLEU-1 looks for matching 1-gram, or words, between the texts. Then, BLEU-2 looks for matching 2-gram, where two words appear sequentially in order in both texts. For example, consider Phrase A, "pretty dog", and Phrase B, "pretty brown dog". Despite "pretty" and "dog" appearing sequentially in both phrases, the BLEU-2 score would be 0 because Phrase B breaks up the 2-gram, "pretty dog", with the word "brown". The case for BLEU-3 and BLEU-4 follows inductively. This process was repeated for each of the four machine translation metrics, whose scores ranged from 0, meaning dissimilar, to 1, meaning very similar. In total, each image had four caption candidates each evaluated by seven machine translation metrics for a total of 28 scores.

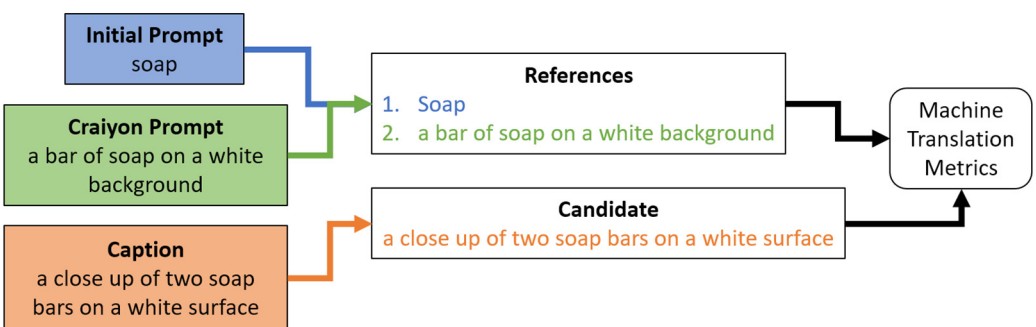

**Figure 4.** Machine translation input transformation.

The cosine similarity metrics are similar, as they range from 0 for dissimilar to 1 for highly similar; however, their implementation is different from the machine translation metrics. Cosine similarity is a popular metric for measuring similarity between vectors, such as word embeddings. However, to apply cosine similarity, it requires that the prompts and captions be transformed into their word embedding form(s), which is model-dependent. Six popular word embedding models were selected: Word2Vec [97,98], GloVe [101], FastText [102], DistilRoBERTa [110], MiniLM-L12 [112], and MiniLM-L6 [112].

Each prompt, the initial and Craiyon versions, and the four generated captions were transformed into their word embedding versions, visually represented in Figure 5. This transformation was performed by retrieving the word embedding for each word present in the prompt/caption from a pre-trained version of the models. In the event the prompt/caption had more than one word, each word in the prompt/caption's embedding was summed to create the overall prompt/caption embedding. Then, the caption embedding was compared separately to the initial prompt embedding and the Craiyon prompt embedding via their cosine similarity (see Equation (1)). In total, each image had its four captions and two prompts compared to one another (eight comparisons) in their six word embedding forms (i.e., eight comparisons by six forms) for a total of 48 cosine similarity scores.

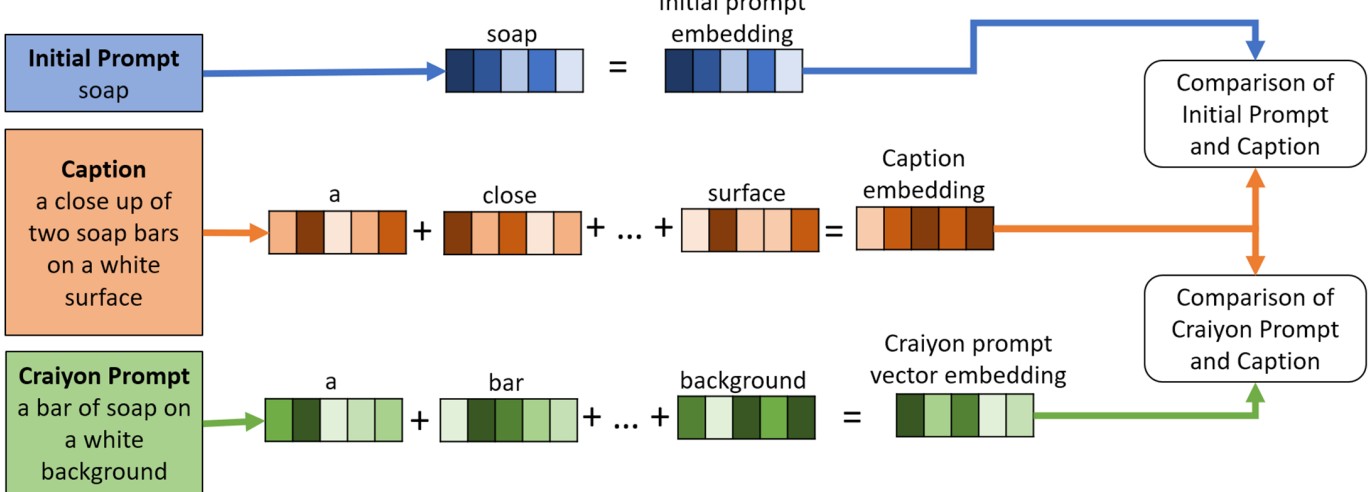

**Figure 5.** Cosine similarity input transformation.

## 4. Results and Discussion

The methodology described in Section 3 was applied to all prompt–caption pairs. An instance of the pipeline we used in this study is shown in Figure 6. An initial prompt is passed to Craiyon, which generates a Craiyon prompt and nine resulting images (for our purposes here, only one of those images is shown). Next, the generated image is passed onto our four image-to-text models, which each generate a caption. Finally, for the evaluation, this one image generates 76 similarity scores. There are 28 machine translation scores representing each of the seven machine translation metrics when evaluating each of the four image-to-text models. The remaining 48 scores are evenly split between those that were comparing the image caption to the initial and the Craiyon prompts. It is notable that Craiyon produces nine images for each prompt; therefore, this is repeated nine times for a total of 684 scores for each properly generated caption.

The results of the average evaluation score for each metric are shown in Tables 5–7. Table 5 shows the metrics for the machine translation methods since the initial and Craiyon prompts were used as references for the candidate (generated caption) to be compared at once. The BLEU, ROUGE, METEOR, and SPICE scores range from 0 (least ideal) to 1 (most ideal). Since this ability was not available for the cosine similarity results, the generated captions' cosine similarities to the initial prompt are shown in Table 6 and their cosine similarity to the Craiyon prompt is shown in Table 7. Due to how cosine similarities are calculated, a negative value is possible, but the effective scale ranges from 0 (completely dissimilar) to 1 (exactly alike). Despite the metrics being measured on the same scale, machine translation scores look at the replication of words, phrases, etc., in the prompts and captions, whereas cosine similarity considers how similar the prompt and caption are to one another.

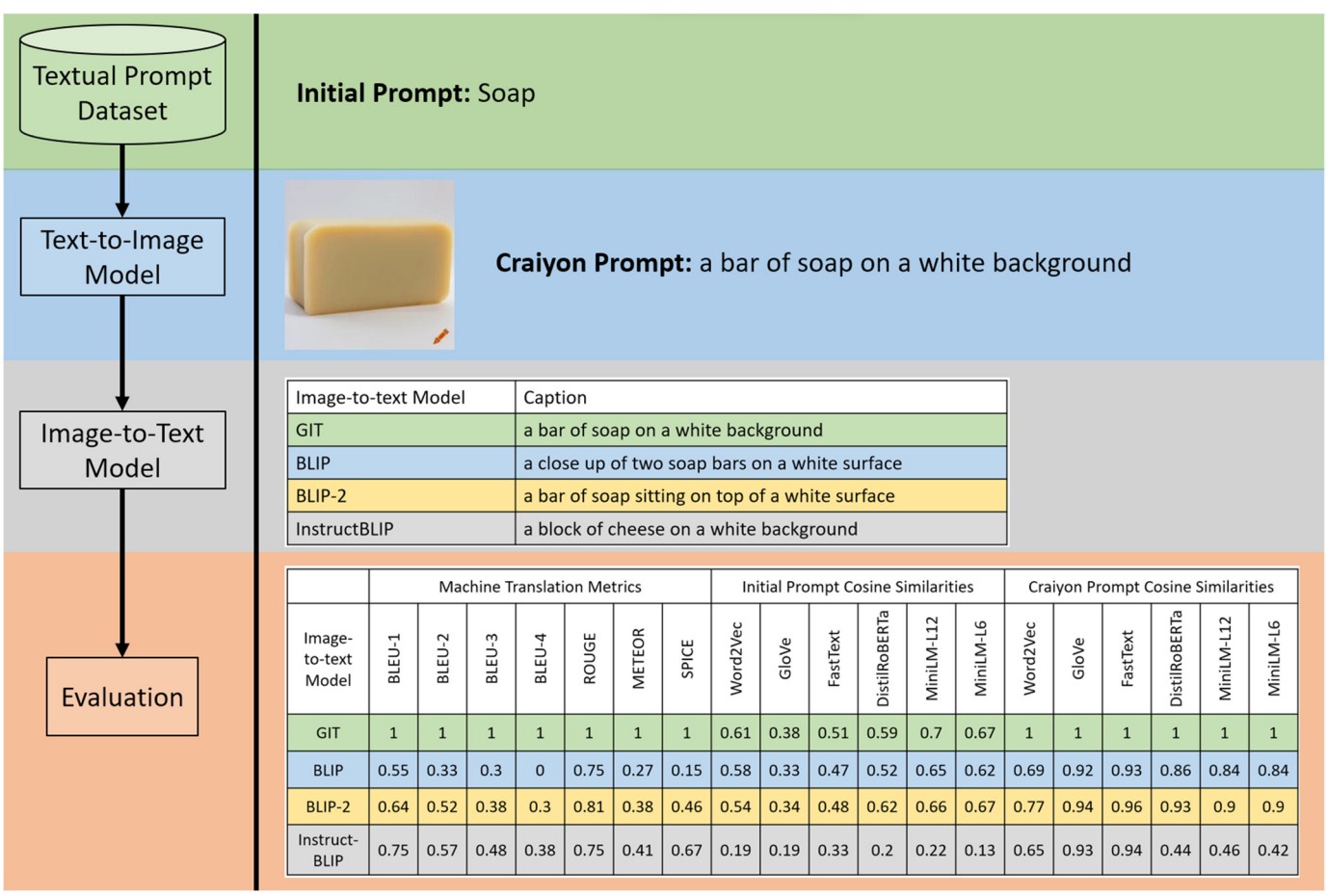

**Figure 6.** Metric calculation example walkthrough.

**Table 5.** Machine translation metrics and scores.

| Model | BLEU-1 | BLEU-2 | BLEU-3 | BLEU-4 | ROUGE | METEOR | SPICE |
|---|---|---|---|---|---|---|---|
| GIT | 19.4% | 4.4% | 1.2% | 0.4% | 23.6% | 9.9% | 7.1% |
| BLIP | 15.1% | 3.4% | 0.8% | 0.2% | 20.3% | 9.3% | 6.6% |
| BLIP-2 | 20% | 4.4% | 1.4% | 0.4% | 24.3% | 10.1% | 7.2% |
| InstructBLIP | 19.4% | 4.5% | 1.4% | 0.5% | 23.5% | 10% | 7.3% |
| Average | 18.5% | 4.2% | 1.2% | 0.4% | 22.9% | 9.8% | 7.2% |

**Table 6.** Average cosine similarity between initial prompt and generated captions.

| Model | Word2Vec | GloVe | FastText | DistilRoBERTa | MiniLM-L12 | MiniLM-L6 |
|---|---|---|---|---|---|---|
| GIT | 32.2% | 40% | 47.7% | 22.3% | 24.7% | 25.3% |
| BLIP | 31.7% | 39.5% | 50.2 | 18.3% | 20.3% | 20.4% |
| BLIP-2 | 32.4% | 40.1% | 48.1% | 21.3% | 23.3% | 24% |
| InstructBLIP | 32.5% | 40.3% | 49.8% | 21.8% | 24% | 24.5% |
| Average | 32.2% | 40% | 49% | 20.9% | 23.1% | 23.6% |

**Table 7.** Average cosine similarity between Craiyon prompt and generated captions.

| Model | Word2Vec | GloVe | FastText | DistilRoBERTa | MiniLM-L12 | MiniLM-L6 |
|---|---|---|---|---|---|---|
| GIT | 41.7% | 72.1% | 78.1% | 28.2% | 27.1% | 28.1% |
| BLIP | 42.5% | 73.8% | 79.2% | 25.5% | 24.3% | 25.3% |
| BLIP-2 | 42.3% | 73.2% | 79.4% | 27.6% | 26.5% | 27.5% |
| InstructBLIP | 43.8% | 72.1% | 78.4% | 28.7% | 27.4% | 28.6% |
| Average | 42.6% | 72.8% | 78.8% | 27.5% | 26.3% | 27.4% |

### 4.1. Machine Translation Results

In Table 5, we see the BLEU-1 score is highest compared to the remaining BLEU scores, which is as expected since the prompts/captions were relatively short (typically less than ten words before the removal of stop words). Given the very small values for BLEU-2 through BLEU-4 (cosine similarity less than 0.05), they may not be appropriate to consider for future similar analyses. ROUGE consistently scored the four image-to-text models the highest, being in the 20–25% range. METEOR scored the captions around the 10% value and SPICE was lower, around the 60–70% range. BLIP consistently scored slightly lower than the remaining three on all the machine translation metrics; however, on average all the metrics scored the prompt–caption comparison relatively low.

### 4.2. Cosine Similarity Results

When using the initial prompt to compare the cosine similarity to the captions in Table 6, we had a wide variety of scores based on the word embeddings from various vector space models (Word2Vec, GloVe, and FastText) and pre-trained language models (DistilRoBERTa, MiniLM-L12, and MiniLM-L6). There is a clear gap of at least 0.05 between the vector space models and the pre-trained language models. Word2Vec scored the captions the lowest of the vector space models, but higher than any of the pre-trained language models, with values around 32%. GloVe scored captions higher, around the 40% similarity mark, but FastText gave the highest similarity scores, near 50% similarity. Though these scores are higher than the machine translation values, 50% would correspond with a neutral prompt/caption, meaning they are neither dissimilar nor similar. All the pre-trained language models gave relatively low similarity scores within the 18–26% range. This performance is expected to a degree since a one-word initial prompt is typically being compared to a multi-word sentence. Using the example from Figure 6, the initial prompt is "soap" and the GIT-created caption is "a bar of soap on a white background". Even though "soap" is in the caption, there are several other words that influence the sentence vector, which would have to cancel one another out perfectly to be left with a vector equivalent to the word embedding for "soap".

Table 7 includes the highest scores from this study across the board. Opposed to comparing the initial prompt, the Craiyon prompt is used for comparison to the generated caption. Since the Craiyon prompts were structured more similarly to the generated captions, it is not surprising that the values in Table 7 are higher than those in Table 6. However, similar behaviors exist between the models presented in both tables; vector space models provide higher similarities to the pre-trained language models. Word2Vec still assigned the lowest similarity scores amongst the vector space models, but there is about a 10% increase from the average similarity using the initial prompt. We see more significant jumps into the 70s for GloVe and FastText, which is approximately a 30% increase from the initial prompt scores in Table 6. All the pre-trained language models remained in the 20–30% range with single-digit increases from when the initial prompt was used.

### 4.3. Major Results

Although Tables 5–7 all have scores within the same range, they cannot necessarily be compared directly to one another given they each have different inputs that affect their resulting output. However, several high-level conclusions can be drawn:

1.  Machine translation methods yielded consistently low scores in comparison to the cosine similarity scores;
2.  For the cosine similarity metrics, the Craiyon prompts yielded higher scores than the initial prompts when comparing them with the generated captions;
3.  Vector space models (Word2Vec, GLoVe, and FastText) were most generous with their similarity scores compared to pre-trained language models;
4.  Image-to-text models minimally affected the similarity scores.

These major results show how various text evaluation methods can be used to evaluate "uncertainty" within generative AI models. As the various image-to-text models did not seem to impact the uncertainty quantification, the evaluation metrics are what researchers should study. The "best" evaluation metric is dependent on validation by human judgment and how much "uncertainty" a human believes to be associated with a prompt–caption pair.

These results suggest there is a significant amount of uncertainty within the AIGC based on our metrics. One potential explanation for this is the existence of many data elements classified as Case B (the prompts do not align with the images) or E (neither the prompts nor the images align with one another) from Figure 3. The detrimental results of this can be seen in Figure 6, where the Craiyon prompt was "a bar of soap on a white background" and the InstructBLIP caption was "a block of cheese on a white background". Though these two sentences have different subjects, they share six out of eight words, yet still received low cosine similarity scores from the pre-trained language models and no score exceeding 0.75 for the machine translation metrics. Minimizing, if not eliminating, any data point that falls into Case B or E (see Figure 3) would assist in ensuring these scores are meaningful.

## 5. Conclusions and Future Work

AI-generated content (AIGC), especially its visual variety, has had an unprecedented rate of production with the rise of high-quality and easy-to-use interfaces exemplified by DALL-E 2, Midjourney, and Craiyon. Despite many astounding results, there are still areas where these generative AI (GAI) models show "uncertainty" when transforming a textual prompt into its corresponding visual counterpart. We first propose a generic pipeline that has four main modules: text-to-image, image-to-text, image quality assessment, and text evaluation methods. The textual prompt dataset is used to prompt the text-to-image generator to create a corresponding image. This image is passed to an image-to-text model to produce a corresponding caption, which is compared to the initial textual prompt used for that particular image.

This generic pipeline was specified in this study such that the textual prompt dataset was the Sternberg and Nigro analogy dataset originally used in [111], but for accessibility, we used its modified version proposed in [113]. The image-to-text model selected was Craiyon V3 due to its cost and versatility [35–37]. Four image-to-text models were selected, which each produced one caption per image: GIT [75], BLIP [76], BLIP-2 [78], and Instruct-BLIP [79]. Several evaluation metrics were selected for comparison, split between typical machine translation metrics (BLEU [89], ROUGE-L [90], METEOR [91], and SPICE [95]) and cosine similarity based on various word embedding models (Word2Vec [97,98], GloVe [101], FastText [102], DistilRoBERTa [110], MiniLM-L12 [112], and MiniLM-L6 [112]). Each evaluation metric was calculated for each prompt–caption pair.

To answer the primary question of how to quantify uncertainty, we used the scores from the evaluation metrics ranging from 0 (dissimiSlar) to 1 (exactly alike). The four image-to-text models behaved comparably to one another across all the metrics used. The machine translation scores on average were lower than the cosine similarity methods, with BLEU-1 scoring the hisghest. There was more variation within the cosine similaritsy metrics. FastText provided the highest similarity scores across all the other metrics; however, GloVe was close behind. Vector space models (Word2Vec, GLoVe, and FastText) appeared to give higher similarity scores compared to the word embedding models (DistilRoBERTa,

MiniLM-12, and MiniLM-6). In conclusion, it appears that the image-to-text model has a limited impact on the analysis, whereas the evaluation metrics differ greatly. The quantification of where AI is certain or uncertain is an important step in the creation of usage guidance and policy.

Regarding future work, one idea would be to eliminate elements of the dataset that fall within Cases B or E to minimize the number of "garbage in, garbage out" results. The ultimate goal is to better engineer the prompts such that the images are always representative of the intended concept. Further exploration into prompt engineering is needed to help eliminate some of these issues and minimize the amount of uncertainty with AIGC. Of the metrics used to evaluate the results, for shorter prompts/captions, as in our case, there is little value added to the BLEU-3 and BLEU-4 scores. These scores may provide more insights when used to evaluate longer prompts/captions. Considering other image-to-text models that provide greater details or longer captions would also be interesting in a later study. Within image quality assessment, a human baseline is often established to which the automated metrics are to be compared in determining which one reflects human judgment the best. A human factors study to establish this quality baseline is currently being conducted by the researchers. Upon the establishment of a baseline, other popular text evaluation metrics may be of interest to explore on the dataset as well.

**Author Contributions:** Conceptualization: K.C., A.M. and T.J.B. Data curation: K.C., A.M. and T.J.B. Methodology: K.C., A.M. and T.J.B. Writing—original draft: K.C., A.M. and T.J.B. Writing—review and editing: K.C., A.M. and T.J.B. All authors have read and agreed to the published version of the manuscript.

**Funding:** This research received no external funding.

**Data Availability Statement:** The modified Sternberg and Nigro dataset from [111] will be made available by the authors on request. The images presented in this article are not readily available because of Department of Defense data and information sharing restrictions.

**Acknowledgments:** The authors would like to thank Arya Gadre and Isaiah Christopherson for generating the image dataset used in this study during the 2023 AFRL Wright Scholars Research Assistance Program. The views expressed in this paper are those of the authors and do not necessarily represent any views of the U.S. Government, U.S. Department of Defense, or U.S. Air Force. This work was cleared for Distribution A: unlimited release under AFRL-2023-5966.

**Conflicts of Interest:** The authors declare no conflicts of interest.

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
