# Peer review of "Uncertainty in Visual Generative AI"

_algorithms, doi:10.3390/a17040136_

Round 1

Reviewer 1 Report

Comments and Suggestions for Authors

This submission has a few editorial corrections that need to be made. I have no corrections or comments about the presentation of the concepts or other of this paper.

The editorial corrections that must be made are the following:

(1.) Page 2 and 3: The paragraph from line 50 on page 2 extends for 27 lines to line 76 and needs to be subdivided into smaller paragraphs to make more comprehensible by the reader.

(2.) Page 4: The paragraph from line 143 to 172 is about 30 lines in length and difficult to comprehend and needs to be subdivided into smaller paragraphs to make more comprehensible by the reader.

(3.) Page 5 top line of Table 2 is missing above text "DALL-E 3 (Quality: HD)".

(4.) Page 11 line 359: "Table 5-7" should be moved left to align with the margin.

(5.) Page 11 lines 370 and 371:  The word "In" needs to  be combined with the text on the following line 371 to read "In Table 5, we see the BLUE-1 ...".

(6.) Page 13 lines 414 and 415; The word "Although" needs to be combined with the text on the following line 415 to read "Although Table 5-7 all have scores within the same range, ...".

(7.) Page 13 line 437 to Page 14 line 467 extends for 31 lines and needs to be sub-divided into smaller paragraphs to improve comprehension by the reader.

(8.) Page 18 Reference 89 is missing capitalization in title for the two words of "annual meeting" that should appear as "Annual Meeting".

(9.) Page 19 Reference 98 is missing capitalization in words in title that should appear as "Proceedings of the 2013 Conference of the North American Chapter of the Association for Computational Linguistics: Human language technologies".

Author Response

Reviewer,

Thank you for taking the time to give a thorough review of our paper, “Uncertainty in Visual Generative AI.” We believe your attention to detail has improved the stylization and readability of our paper. See our responses below.

  1. On pages 2-3, the larger paragraph has been split into three smaller paragraphs (spanning lines 50-77) in accordance with natural breaks in the content.
  2. On page 4, the larger paragraph has been split into six smaller paragraphs (spanning lines 144-176) in accordance with natural breaks in the content.
  3. On page 5, the right-most column’s header title, “Model”, has been added.
  4. On page 12, “Tables 5-7” now aligns with the margin (now line 417). The original alignment issue likely arose due to the alignment of the table caption.
  5. On page 12, “In Table 5…” has been fixed (now line 430). The original alignment issue likely arose due to the alignment of the table caption.
  6. On page 14, “Although Tables 5-7…” has been fixed (now line 473). The original alignment issue likely arose due to the alignment of the table caption.
  7. On pages 14-15, the larger paragraph has been split into three smaller paragraphs (spanning lines 501-532) in accordance with natural breaks in the content.
  8. On page 20, reference 89 has been corrected such that “Annual Meeting” is capitalized.
  9. On page 20, reference 98 has been corrected such that the name of the proceedings is properly capitalized.

Per the other reviews, additional information has been added to the Methodology (Section 3) and Results & Discussions (Section 4) to improve the clarity and validity of our approach and the results thereof. All corresponding changes are highlighted in yellow in the revised version.

Reviewer 2 Report

Comments and Suggestions for Authors

The authors propose a general pipeline to automatically quantify uncertainty within GAI. This paper proposed the processes of text-to-image, image-to-text, quality assessment of image and text evaluation methods to overcome the uncertainty in transformation of textual prompt into its corresponding images. In order to publish this paper in this journal, the authors must revise and supplement the following problems.

(1) I believe that the authors wrote this paper in a hurry. Overall, authors will need to make several changes to the format of the paper, including line alignment and section numbers. For examples,

3.3. Textual Evaluation Metrics -> 3.4. Textual Evaluation Metrics

6. Conclusions & Future Work -> 5. Conclusions & Future Work

(2) The authors showed a specific process related to their research in section 3. But, to clearly understand their research method, the authors have to describe more general architecture of their proposed method in section 3.

(3) In the experimental results in section 4, I had difficulty confirming the validity of the proposed method. More clear explanations (additional experiments, if necessary) are needed to understand the performance of the proposed method.

Round 2

Reviewer 2 Report

Comments and Suggestions for Authors

The authors resolved all issues raised by the review results.